# A direct coupled electrochemical system for capture and conversion of $CO_2$ from oceanwater

Ibadillah A. Digdaya [1], Ian Sullivan [1], Meng Lin [2✉], Lihao Han[1], Wen-Hui Cheng [3],
Harry A. Atwater [3✉] & Chengxiang Xiang[1✉]

Capture and conversion of $CO_2$ from oceanwater can lead to net-negative emissions and can provide carbon source for synthetic fuels and chemical feedstocks at the gigaton per year scale. Here, we report a direct coupled, proof-of-concept electrochemical system that uses a bipolar membrane electrodialysis (BPMED) cell and a vapor-fed $CO_2$ reduction ($CO_2$R) cell to capture and convert $CO_2$ from oceanwater. The BPMED cell replaces the commonly used water-splitting reaction with one-electron, reversible redox couples at the electrodes and demonstrates the ability to capture $CO_2$ at an electrochemical energy consumption of 155.4 kJ mol$^{-1}$ or 0.98 kWh kg$^{-1}$ of $CO_2$ and a $CO_2$ capture efficiency of 71%. The direct coupled, vapor-fed $CO_2$R cell yields a total Faradaic efficiency of up to 95% for electrochemical $CO_2$ reduction to CO. The proof-of-concept system provides a unique technological pathway for $CO_2$ capture and conversion from oceanwater with only electrochemical processes.

[1] Joint Center for Artificial Photosynthesis and Division of Chemistry and Chemical Engineering, California Institute of Technology, Pasadena, CA 91125, USA. [2] Department of Mechanical and Energy Engineering, Southern University of Science and Technology, 518055 Shenzhen, China. [3] Joint Center for Artificial Photosynthesis and Department of Applied Physics and Materials Science, California Institute of Technology, Pasadena, CA 91125, USA.
✉email: linm@sustech.edu.cn; haa@caltech.edu; cxx@caltech.edu

Capture and conversion of $CO_2$ from anthropogenic emission is becoming an increasingly important social responsibility as the concentration of atmospheric $CO_2$ continues to rise above record high levels[1,2]. $CO_2$ from the atmosphere, oceanwater and point sources are considered as the major feedstock for subsequent capture and conversion processes[3]. Since the 1000 largest power plants emit >20% of total global fossil fuel $CO_2$ emissions[4], capture of $CO_2$ from point sources, e.g., flue gas, which often contain higher $CO_2$ concentrations (10%), has been the focus in the carbon capture and sequestration approach[5,6]. However, to achieve negative emissions in the long term, capture $CO_2$ directly from air[7–14] or oceanwater[15–19] will likely play a much bigger role[20–23]. World ocean constitutes the largest carbon sink, absorbing about 40% of anthropogenic $CO_2$ since the beginning of industrial era[24–26] with an effective $CO_2$ concentration of 2.1 mmol kg$^{-1}$, or 0.095 kg m$^{-3}$ in oceanwater, which is a factor of 120 times larger than in the atmosphere[27–29]. Thus, $CO_2$ capture from oceanwater provides an alternative and unique approach to direct air capture (DAC) in the global carbon removal technological landscape[30]. $CO_2$ capture from oceanwater, however, presents many challenges. For example, the estimated cost of oceanwater intake, pre-treatment and outfall in a land-based, stand-alone system is high, ~$1.40 kg$^{-1}$ $CO_2$[19]. While co-location with a desalination plant could reduce this cost[19], the system scale for $CO_2$ removal would be limited to less than 100 kt-$CO_2$ year$^{-1}$ based on the current largest desalination plant[31]. Development of an off-shore, stand-alone system powered by renewables can alleviate competitive land use, allow unique access to off-shore $CO_2$ storage sites, and can provide a source of $CO_2$ for off-shore enhanced oil recovery.

Until now, two types of electrodialysis designs have been reported for $CO_2$ capture from oceanwater[15–17]. The basic operating principle of the electrodialysis for $CO_2$ capture is to push the $CO_2$/bicarbonate equilibrium toward dissolved $CO_2$ by acidifying the oceanwater. The acidified stream is then passed through a liquid–gas membrane contactor, which captures the gaseous $CO_2$ from the dissolved $CO_2$ in the aqueous stream. The unintended water-splitting reaction, i.e., hydrogen evolution reaction (HER) at the cathode and oxygen evolution reaction (OER) at the anode, in the previously reported devices[15,17] often resulted in additional voltage loss and additional electrochemical energy consumption for $CO_2$ removal. Herein, a new bipolar membrane electrodialysis (BPMED) cell design, in which the HER and OER at the electrodes is replaced with reversible redox-

couple reactions with minimal thermodynamic and kinetic voltage losses, is designed, constructed, and evaluated. At an operating current density of 3.3 mA cm$^{-2}$, and an oceanwater flow rate of 37 ml min$^{-1}$, a record low electrochemical energy consumption of 155.4 kJ mol$^{-1}$ or 0.98 kWh kg$^{-1}$ of $CO_2$ is achieved. By contrast, the thermodynamic limit of electrochemical energy consumption for converting $CO_2$ into fuels is much larger. For example, converting $CO_2$ to methane requires 13.9 kWh kg$^{-1}$ of $CO_2$. Hence the electrochemical energy required for $CO_2$ capture from oceanwater would only constitute a small fraction of the total capture and conversion energy. The demonstrated BPMED cell also exhibits a high $CO_2$ capture efficiency of 71% of the total dissolved inorganic carbon (DIC). In addition, we demonstrate the direct coupling between the $CO_2$ capture from oceanwater via BPMED and electrochemical $CO_2$ reduction ($CO_2$R) into fuels and chemicals. The vapor-fed $CO_2$R cell converts $CO_2$ from oceanwater to fuels and chemicals such as carbon monoxide, ethylene, ethanol, and propanol with total Faradaic efficiency (FE) of up to 73% at current densities of 58 mA cm$^{-2}$ using Cu electrocatalyst and to CO with FE of up to 95% at current densities of 11.15 mA cm$^{-2}$ using Ag electrocatalyst.

## Results

### Design and fabrication of the BPMED cell for $CO_2$ capture from oceanwater.
Figure 1a shows the schematic illustration of the BPMED cell for $CO_2$ capture from oceanwater. The BPMED cell contained two oceanwater compartments separated by a bipolar membrane (BPM), two reversible redox-couple compartments, each separated from the oceanwater compartment by a cation exchange membrane (CEM), and two electrodes for electrochemical reactions. The electrochemical reactions at the electrodes, ionic transport across the membranes, and water dissociation at the BPM interface are illustrated in Fig. 1a. At the middle of the BPMED cell, a BPM that generates proton ($H^+$) and hydroxide ion ($OH^-$) fluxes via water dissociation reactions at the BPM interface was used to convert the input oceanwater into output streams of acidified and basified oceanwater. The electrode solution, i.e., catholyte and anolyte, contained a reversible redox-couple solution, potassium ferro/ferricyanide ($K_3$/$K_4$[Fe(CN)$_6$)]), and was re-circulated to minimize any polarization losses associated with concentration overpotentials at the electrodes. Two CEMs were then employed to charge balance the acidified or basified streams of oceanwater by selectively transporting cations from the anolyte or toward the catholyte,

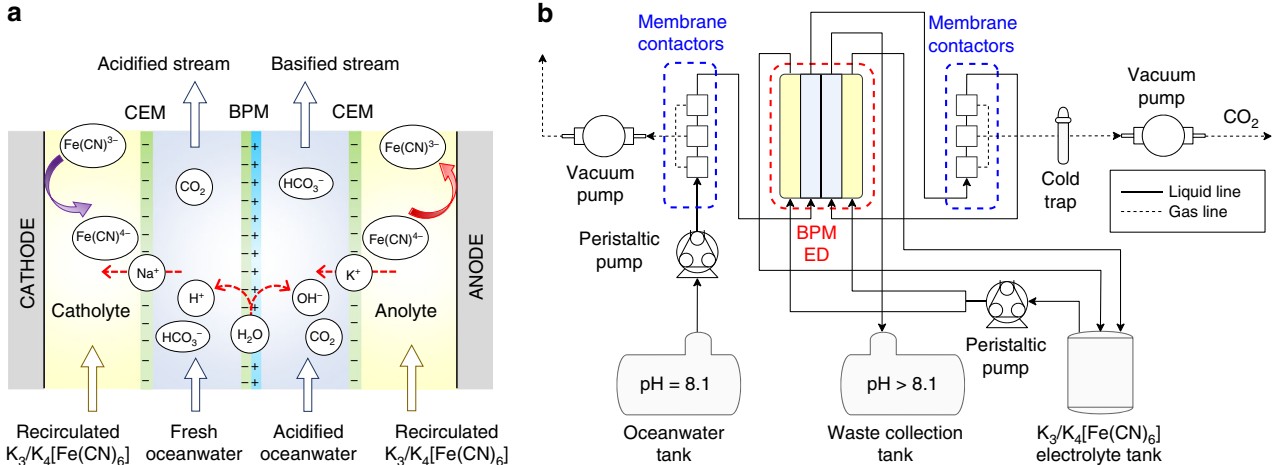

**Fig. 1 BPM electrodialysis and $CO_2$ capture system. a** Schematic illustration of the BPM electrodialysis cell. **b** Process flow diagram of the experimental setup for $CO_2$ capture from oceanwater.

respectively. The electrode reactions in the BPMED cell were one electron, reversible redox reaction as the following:

$$\text{Cathode}: \left[\text{Fe}(\text{CN})_6\right]^{3-} + e^- \rightarrow \left[\text{Fe}(\text{CN})_6\right]^{4-}. \tag{1}$$

$$\text{Anode}: \left[\text{Fe}(\text{CN})_6\right]^{4-} \rightarrow \left[\text{Fe}(\text{CN})_6\right]^{3-} + e^-. \tag{2}$$

One unique advantage of this new BPMED configuration is that it can be employed and scaled up both in a single stack configuration or a multi-stack configuration without introduction of any unintended chemical reactions or any additional voltage losses. By contrast, the BPMED configuration in Supplementary Fig. 1a[17] can only be employed in a single stack configuration with a untunable ratio of the $CO_2$ capture rate and the $H_2$ generation rate, while BPMED configuration as shown in Supplementary Fig. 1b[15] can only be employed in a multi-stack configuration to minimize the voltage penalty associated with water-splitting reactions.

Figure 1b shows the experimental flow diagram of the electrochemical capture and conversion of $CO_2$ from oceanwater. Dissolved gasses in the input oceanwater stream, e.g., $O_2$ and $N_2$, were vacuum stripped using three commercial membrane contactors (3M™ Liqui-Cel™ MM-0.5 × 1, each with a maximum operating liquid flow rate of 30 ml min$^{-1}$) connected in a series prior to entering the acidification compartment. The acidified oceanwater was directed toward another series of three membrane contactors for removal of dissolved $CO_2$ by a vacuum pump. A cold trap surrounded by dry ice was used to condense moisture from the gas output. The acidified oceanwater was then fed to the base compartment where the pH was retrieved close to the initial value and the effluent was disposed of as a waste in a collection tank.

**Polarization losses in the BPMED cell.** The electrochemical performance of the BPM is the key to the operation of BPMED cell, and to understand the voltage loss across the BPM, a multi-physics model was used to simulate the voltage–current density characteristics, electrochemical potentials, and partial current densities carried by different ions in the system (Supplementary Note 1). Figure 2a indicates that the simulated and the experimental data showed good agreement throughout the entire current density range. The total current density ($j_{total}$) equals to the sum of the current density carried by the hydrogen ion ($j_{H^+}$), the hydroxide ion ($j_{OH^-}$), and the sodium and chloride co-ions ($j_{Na^+}$ and $j_{Cl^-}$) in the solution. Other counter ions also co-exist in the oceanwater but their concentration was too small to impact the total current density. The complete list of anions and cations in the synthetic oceanwater is provided in Supplementary Table 4. At low current densities, the leak current of $Na^+$ and $Cl^-$ was substantial due to the imperfect permselectivity[32,33] of the cation exchange layer (CEL) and anion exchange layer (AEL) of the BPM. When the BPM voltage exceeded 0.4 V, the water dissociation reaction started to take place due to the increased electric field across the BPM interface, and at large BPM voltages, the water dissociation rate, i.e., $j_{H^+}$ and $j_{OH^-}$, became the dominating partial current density in the system. In this study, the operating current density of the device was set to >3.3 mA cm$^{-2}$ and from Fig. 2b, >93% of the ionic transport was carried by $j_{H^+}$ and $j_{OH^-}$ with minimal contribution from co-ion crossovers.

Figure 2c shows the total cell voltage as a function of the operating current density using different electrode solutions and flow conditions. The thermodynamic limit of the total cell voltage ($V_{cell,\ ideal}$) in the BPMED cell can be expressed in the following equation:

$$V_{cell,\ ideal} = \frac{RT}{F}\left(\text{pH}_{basified} - \text{pH}_{acidified}\right), \tag{3}$$

where $R$ is the universal gas constant (8.3144 J K$^{-1}$ mol$^{-1}$), $T$ is the temperature, $F$ is the Faraday constant (9.6485 × 10$^4$ C mol$^{-1}$), pH$_{basified}$ is the pH of the solution in the basified compartment and pH$_{acidified}$ the pH of the solution in the acidified compartment. In comparison, the practical total cell voltage ($V_{cell,\ practical}$) can be expressed as the following equation:

$$V_{cell,\ practical} = \frac{RT}{F}\left(\text{pH}_{basified} - \text{pH}_{acidified}\right) + V_{BPM\ loss} + V_{CEMs}$$
$$+ V_{oceanwater} + V_{electrolyte} + V_{electrode}, \tag{4}$$

where $V_{BPM\ loss}$ is the voltage loss across the BPM, $V_{CEMs}$, is the voltage loss across the CEMs, $V_{oceanwater}$ and $V_{electrolyte}$ are the voltage loss across the oceanwater and electrolyte compartment, respectively, and $V_{electrode}$ is the voltage loss at the two electrodes. The dominating voltage penalty in the BPM-based electrodialysis cell originated from the water dissociation kinetics and polarization loss across the BPM. As shown in Fig. 2c, the voltage of the electrodialysis cell with the traditional 0.5 M $Na_2SO_4$ electrode solution was significantly higher than any of the redox-couple-based cell configurations in all current density ranges due to the required thermodynamic voltage window (1.23 V) for water-splitting as well as kinetic overpotentials for OER and HER. Eliminating the water-splitting reaction in the BPM-based electrodialysis cell by replacing the traditional electrode electrolyte with $K_3/K_4[\text{Fe}(\text{CN})_6]$ redox-couple solutions significantly reduced $V_{electrode}$ and hence reduced the total operating cell voltage.

Figure 2c also indicates that polarization losses associated with concentration overpotentials in the redox-couple compartments can be minimized by increasing the concentration and flow rates of the redox-couple solutions. The total cell voltage was very close to the voltage difference across the BPM at a concentration of 0.4 M and a flow rate of 40 ml min$^{-1}$, particularly at the low current density regime. The main difference between $V_{cell,\ practical}$ and $V_{cell,\ ideal}$ was $V_{BPM\ loss}$ when the rest of the voltage losses were minimized by the optimized cell design. In the linear region of voltage–current density curves, the discrepancy between the BPM voltage and cell voltage was primarily due to the resistance of the CEM and the 0.4 M $K_3/K_4[\text{Fe}(\text{CN})_6]$ solution. The dominating voltage penalty in the BPM-based electrodialysis cell originated from the water dissociation kinetics and polarization loss across within the BPM. Similar or higher voltage losses across the BPM during operation were observed in previous reports[32–35].

The calculated and the experimentally measured pH (Fig. 2d) as a function of the current density in the BPM electrodialysis cell showed excellent agreement throughout the whole range of applied current density. The contributions of co-ions transport, e.g., $j_{Na^+}$ and $j_{Cl^-}$, to the total current density were significant when the operating current density was lower than 0.4 mA cm$^{-2}$. The water dissociation reaction, i.e., $j_{H^+}$ and $j_{OH^-}$, was the major charge carrier during the process at higher current densities. As shown in Fig. 2d, to attain the desired pHs at the acidified stream, the required current densities were higher for oceanwater that flowed at higher rates. The solution pH has a significant impact on the concentration of the dissolved $CO_2$ and hence on the capture efficiency of the BPM electrodialysis cell. In the synthetic oceanwater, the concentration of the dissolved $CO_2$ increases from 0.016 to 3.08 mM when the solution pH decreases from 8.1 to 4 (Supplementary Fig. 4). Therefore, to efficiently capture $CO_2$ from oceanwater, the solution pH needed to be kept close to 4. Note that the pH of the acidified stream was dictated solely by the operating current normalized with the volumetric flow rate of the oceanwater (Supplementary Fig. 6). All the reported electrodialysis cells at near optimal operating conditions yielded a normalized operating current of 5.71 mA min ml$^{-1}$.

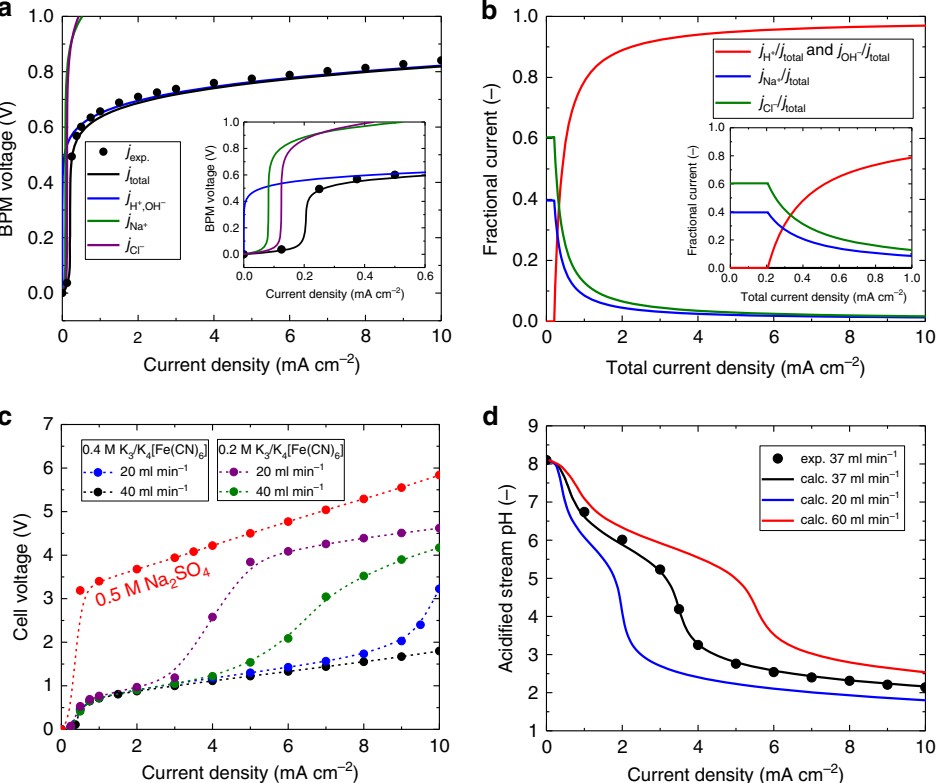

**Fig. 2 BPM electrodialysis performance characteristics. a** Simulated BPM voltage vs. total current density (black line) as well as partial current density carried by major ions, including proton and hydroxide for water dissociation (blue line), $Na^+$ (green line) and $Cl^-$ (purple line), in the synthetic oceanwater. Black dots indicate the experimentally measured BPM voltage as a function of the applied current density, corrected for the ohmic resistance in the oceanwater. **b** Fractional current carried by $H^+$ $(j_{H^+}/j_{total})$, $OH^-$ $(j_{OH^-}/j_{total})$, $Na^+$ $(j_{Na^+}/j_{total})$ and $Cl^-$ $(j_{Cl^-}/j_{total})$ across the BPM as a function of the total current density $(j_{total})$. **c** Experimentally measured voltage-current density characteristics of the BPM electrodialysis cell using 0.5 M $Na_2SO_4$ electrode solution at a flow rate of 40 ml min$^{-1}$ (red dots), 0.4 M $K_3/K_4[Fe(CN)_6]$ at a flow rate of 40 ml min$^{-1}$ (black dots) and 20 ml min$^{-1}$ (blue dots), 0.2 M $K_3/K_4[Fe(CN)_6]$ at a flow rate of 40 ml min$^{-1}$ (green dots) and 20 ml min$^{-1}$ (purple dots). Dashed lines are used to guide the eye. **d** Experimentally measured (black dots) and calculated (lines) pH of the acidified stream as a function of applied current density for BPM electrodialysis cell with an electrode and membrane active area of 64 cm$^2$.

During the experiment, the net ion movement between the electrolyte and the oceanwater replaced the $K^+$ ions with the $Na^+$ ions in the electrolyte, resulting $Na_3$–/$Na_4$–Fe(CN)$_6$ solution in the electrode compartment that is free of $K^+$ after a period of operation. The exchange of the cations in the redox-couple species did not impact the performance of BPMED. Because of the much higher concentration of $Na^+$ (~0.4 M) in oceanwater relative to $H^+$ at mild pH ~5 ($10^{-5}$ M), the transference number was close to unity for $Na^+$ across the CEM, and the transport of $H^+$ to the electrolyte was negligible. In addition, the catholyte and anolyte were circulated during the operation. As a result, the change of the catholyte or anolyte pH was not observed under operating conditions. However, it is important to maintain the acidified oceanwater compartment at mild pHs so that minimal $H^+$ transfer took place between the oceanwater and the catholyte.

**Performances of the BPM-based electrodialysis cell for $CO_2$ capture from oceanwater.** One critical metric for evaluating the performance of the BPMED cell for $CO_2$ capture from oceanwater is the electrochemical energy consumption via electrodialysis. The electrochemical energy consumption is defined as the amount of electrical energy required (in kilowatt hour, kWh) for electrodialysis divided by the amount of captured $CO_2$ (in mass, kg). Figure 3a shows the calculated electrochemical energy consumption as a

function of the applied current density and the oceanwater flow rate. The experimentally measured voltage–current density characteristics of the cell with the optimized $K_3/K_4[Fe(CN)_6]$ electrode solution (Fig. 2c), the calculated pH-current density relations (Fig. 2d), and the $CO_2$ concentration-pH equilibrium (Supplementary Note 2) were used to determine the electrical power consumption and the resulting dissolved $CO_2$ in the oceanwater that can be captured using a traditional liquid–gas membrane contactor. Detailed calculation and flowchart outlining calculation steps are provided in Supplementary Note 3 and Supplementary Fig. 7, respectively. In this calculation, current densities greater than 0.4 mA cm$^{-2}$ were used, where the water dissociation at the BPM interface dominated the ionic transports (Fig. 2b). As shown in Fig. 3a, at any given oceanwater flow rate, there is an optimal operating current density of the cell that would yield the lowest electrochemical energy consumption for $CO_2$ capture. Improving the water dissociation kinetics at the BPM interface as well as lowering the series resistance at high current densities by improving the cell design would further lower electrochemical energy consumption for $CO_2$ capture from oceanwater.

Another important parameter for the BPM-based electrodialysis device is the output rate of the captured $CO_2$. Figure 3b shows the calculated rate of ideal $CO_2$ output as a function of the applied current density and the oceanwater flow rate. The ideal $CO_2$ output rate assumes that all the dissolved $CO_2$ can be

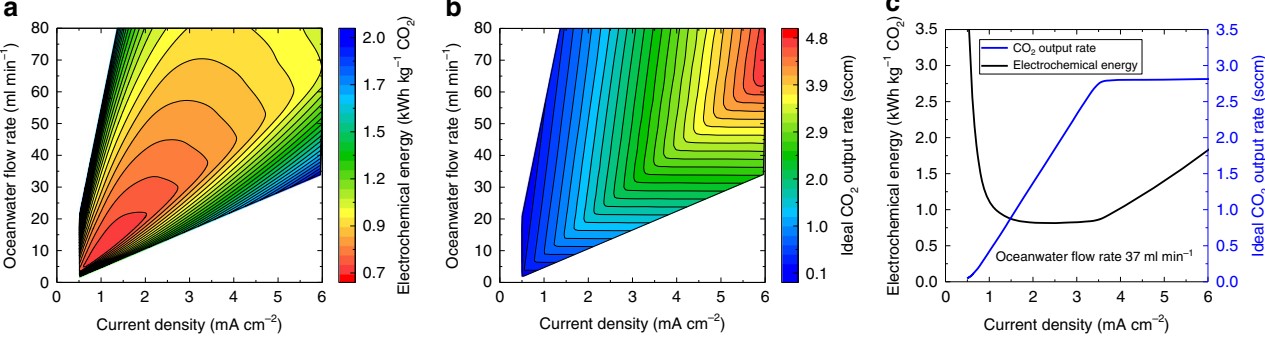

**Fig. 3 Optimization of BPM electrodialysis operating conditions. a** Contour plots of the calculated electrochemical energy consumption for $CO_2$ capture (kWh kg$^{-1}$) and **b** the calculated rate of ideal $CO_2$ output from the BPM electrodialysis cell as a function of the applied current density and the input oceanwater flow rate. The BPM electrodialysis cell has an active electrode and membrane area of 64 cm$^2$. **c** The calculated electrochemical energy consumption and the ideal $CO_2$ output rate as a function of the operating current density at an oceanwater flow rate of 37 ml min$^{-1}$.

captured from the acidified oceanwater using membrane contactors. At any given oceanwater flow rate, the rate of $CO_2$ output increased as the operating current density of the cell increased until a maximum rate for $CO_2$ capture was reached. At a higher oceanwater flow rate, a higher operating current density was required to acidify the oceanwater to the pH at which most carbonate and bicarbonate ions were converted to dissolved $CO_2$. Figure 3c indicates that, at a given oceanwater flow rate of 37 ml min$^{-1}$, there is an optimal regime of operating current density between ~1.5 and ~3.5 mA cm$^{-2}$, where the lowest electrochemical energy consumption was achieved. At the lower operating current density, the pH of oceanwater stream was not sufficiently low to convert the majority of DIC into dissolved $CO_2$, which resulted in low capture efficiency and high electrochemical energy consumption. At higher operating current densities, while the oceanwater stream was sufficiently acidified, the increased voltage across the BPMED led to increased electrochemical energy consumption. The output $CO_2$ flow rate increased linearly as a function of the operating current density until a turning point was reached, where the vast majority of DIC was converted into dissolved $CO_2$. As a result, at an oceanwater flow rate of 37 ml min$^{-1}$, the operating current density was set to 3.3 mA cm$^{-2}$ to minimize the electrochemical energy consumption, while maximizing the output $CO_2$ flow rate.

Figure 4a shows the experimentally measured $CO_2$ capture rate in the BPMED cell, at an applied current density of 3.3 mA cm$^{-2}$ (or an absolute current of 211.2 mA) and at an oceanwater flow rate of 37 ml min$^{-1}$. At these operating conditions, the pH of the acidified stream was 4.7, the measured total voltage across the cell was at ~1 V (Supplementary Fig. 8), the DIC rate was calculated to be 2.8 sccm and the ideal $CO_2$ output rate was 2.6 sccm (Supplementary Note 3). The initial total gas output was ~3 sccm, and stabilized at 2.1 sccm after 1 h of operation when the vacuum pressure was sufficiently low and the remaining air in the gas stream line (e.g., membrane contactors, cold trap, tubing) had been completely evacuated.

To accurately quantify the $CO_2$ concentration, the output gas stream was diluted with pure $N_2$ gas in a mixing chamber before introducing to a gas chromatograph (GC). Figure 4b indicates that during the first hour of the experiment, the output $CO_2$ gradually increased to a constant concentration of 93%, while $O_2$ and $N_2$ decreased to a stable concentration of 1.5% and 5.5%, respectively. The initially low concentration of $CO_2$ was attributed to the incomplete removal of dissolved air from the input oceanwater and the incomplete evacuation of air from the gas stream line. After 1 h of operation, the remaining $O_2$ and $N_2$ measured in the output gas were likely from the slight leak in the

membrane contactors because the ratio between $O_2$ and $N_2$ was roughly 1:4. In the absence of a vacuum stripping stage prior to the BPMED unit, the captured gas contained more than 30% of $N_2$ and $O_2$ gas mixtures (Supplementary Fig. 9).

Figure 4c shows the $CO_2$ capture efficiency and the membrane contactor efficiency over the course of 2 h. The $CO_2$ capture efficiency is defined as the measured rate of the gaseous $CO_2$ output divided by the rate of total DIC in the oceanwater that was introduced into the BPMED cell. The total DIC in the synthetic oceanwater was 3.12 mM (Supplementary Note 2). The membrane contactor efficiency is defined as the measured rate of the gaseous $CO_2$ divided by the rate of dissolved $CO_2$ gas present in the acidified stream at the corresponding pH and the oceanwater flow rate. After the output gas reached a stable rate and composition, the $CO_2$ capture efficiency and the membrane contactor efficiency were 71% and 76%, respectively. The small discrepancy between the $CO_2$ capture efficiency and the membrane contactor efficiency suggests that at these operating conditions (at an applied current density of 3.3 mA cm$^{-2}$, or an absolute current of 211.2 mA and at an oceanwater flow rate of 37 ml min$^{-1}$), most of the DIC in the oceanwater was converted to dissolved $CO_2$. It is important to note that three membrane contactors in series (3M™ Liqui-Cel™ MM-0.5×1) were used at the inlet (for dissolved gas removal, such as $O_2$ and $N_2$, from fresh oceanwater) and the outlet (for $CO_2$ capture) of the acid compartment of the BPMED cell, each with a maximum operating liquid flow rate of 30 ml min$^{-1}$, to separate dissolved $CO_2$ from oceanwater at efficiencies reported herein. Using only 1 membrane contactor also allowed for the separation and removal of $CO_2$, but at a lower efficiency (Supplementary Fig. 10), and two series of membrane contactors were at least required to capture $CO_2$ at efficiencies of more than 70% (Supplementary Fig. 11). The highest experimentally recorded $CO_2$ capture efficiency in this study was 77% at an oceanwater pH of 3.7, where ~99% of the DIC was converted to dissolved $CO_2$ (Supplementary Fig. 12). Adding more membrane contactors may increase the air leakage into the membrane contactors and will not significantly improve the capture efficiency because the maximum removal of dissolved $CO_2$ by vacuum stripping through this type of contactor is typically ~80%[15], which is also consistent with the specification provided by the manufacturer. Figure 4d shows the electrochemical energy consumption for $CO_2$ capture in kWh kg$^{-1}$ and kJ mol$^{-1}$. After 1 h of operation, the electrochemical energy consumption was stabilized to 0.98 kWh kg$^{-1}$ or 155.4 kJ mol$^{-1}$ of $CO_2$. The low electrochemical energy consumption was achieved by eliminating the voltage loss at the electrodes.

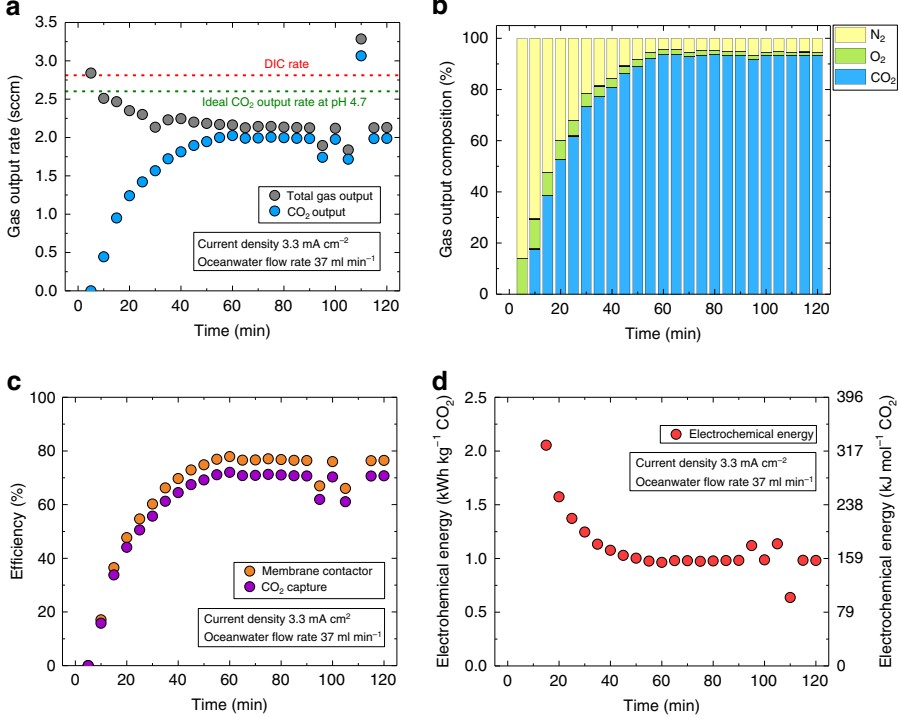

**Fig. 4 CO$_2$ capture performance.** CO$_2$ capture performance as a function of time at an operating current density of 3.3 mA cm$^{-2}$ and an oceanwater flow rate of 37 ml min$^{-1}$ (pH 4.7). **a** The experimentally measured total gas output rate and the captured CO$_2$ output flow rate. The red dashed line indicates the DIC rate at the given oceanwater flow rate and the green dashed line indicates the ideal rate of CO$_2$ output at these operating conditions. **b** The output gas composition. **c** The CO$_2$ capture and membrane contactor efficiency. **d** The electrochemical energy consumption.

In order to sustain the CO$_2$ capture from oceanwater at scale, the oceanwater waste from the BPMED should be returned to the ocean at restored alkalinity to allow for continuous uptake of atmospheric CO$_2$. The removal of CO$_2$ in the oceanwater allows the carbonate species to re-equilibrate and prompts the ocean-water pH to adjust according to the new equilibrium condition. At an oceanwater flow rate of 37 ml min$^{-1}$, a current density of 3.3 mA cm$^{-2}$, the pH of the acidified stream was 4.7, and increased to 5.3 after CO$_2$ capture at an efficiency of 71%. According to the rate of hydroxide generation in the base compartment, which is equal to the rate of proton generation in the acid compartment, the output of oceanwater in the basified stream should have a pH of 10.46. However, experimental measurements showed a pH of 8.5, close to the original oceanwater pH. This discrepancy was attributed to the presence of non-negligible amounts of Mg$^{2+}$ and Ca$^{2+}$ ions in the oceanwater (Supplementary Table 4), which preferentially reacted with OH$^-$ and formed white precipitates of divalent hydroxides and carbonates, as observed during the experiments. In the absence of Mg$^{2+}$ and Ca$^{2+}$, the basified stream would reach a pH of ~10.46 (Supplementary Note 4). Using simplified oceanwater without Mg$^{2+}$ or Ca$^{2+}$, the measured pHs from the acidified stream and basified stream showed good agreement with the calculated values. By contrast, when synthetic oceanwater that contained Mg$^{2+}$ and Ca$^{2+}$ was used, the basified stream exhibited smaller pH increase than the calculated value due to the preferential reaction between OH$^-$ and Mg$^{2+}$/Ca$^{2+}$ (Supplementary Fig. 13). Softening the oceanwater feed prior to BPMED will solve this issue and restore the oceanwater alkalinity but the environmental impacts of returning decarbonized oceanwater at pH >10 with the same salt level is not well understood presently. Subsequent processes will need to be developed and implemented to levitate any impact on oceanic life.

The operating parameters of the proof-of-concept device was constrained by the laboratory hardware. As a result, at an oceanwater flow rate of 37 ml min$^{-1}$, the operating current density was set to 3.3 mA cm$^{-2}$ to minimize the electrochemical energy consumption, while maximizing the CO$_2$ output rate. In a scaled-up device, a much higher oceanwater feed rate would require a higher operating current density to achieve the optimal pH in the acidified compartment and to capture the majority of the CO$_2$. The higher operating current density in a practical device reduces the BPM cost per kilogram of CO$_2$ captured in the system but increases the electrochemical energy consumption due to the increased polarization loss. The trade-off between the electrochemical energy cost and the membrane cost in the overall capture cost of CO$_2$ in the BPMED system at different current densities was analyzed in Supplementary Note 5.

**Electrochemical conversion of CO$_2$ in a vapor-fed device.** Vapor-fed CO$_2$R cells have several advantages for CO$_2$R such as the ability to overcome mass transport limitations of CO$_2$ solu-bility in aqueous electrolytes[36]. Examples of vapor-fed CO$_2$R cells have exhibited large current densities, increased selectivity, and high single pass conversion rates for CO$_2$R on Cu-based electro-des[37–42]. To test the proof-of-concept design, a vapor-fed CO$_2$R cell similar to other previously reported cells was employed[38]. The outlet CO$_2$ stream from the BPMED cell was directly fed through tandem vapor-fed cells, the first for oxygen reduction reaction (ORR) and second for CO$_2$R. The first ORR pre-electrolysis cell was used to eliminate any residue O$_2$ from flowing into the vapor-fed CO$_2$R cell (Supplementary Fig. 17), as any O$_2$ will be selec-tively reduced and lower the FE of the CO$_2$R cell. Two types of CO$_2$R catalysts were used in this study. The Cu catalyst was deposited on a gas diffusion layer (GDL, Sterlitech PTFE) by

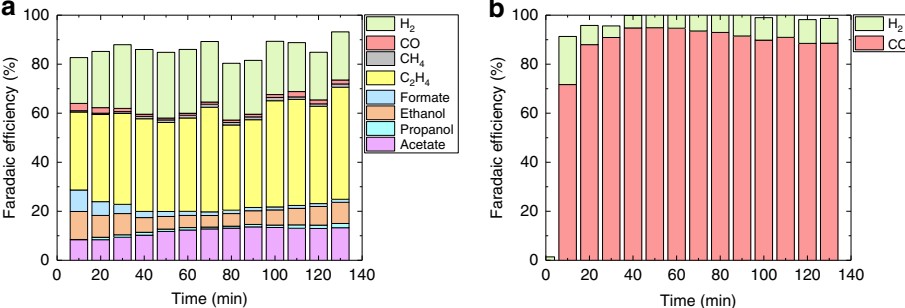

**Fig. 5 Electrochemical conversion of $CO_2$ from oceanwater.** Gas and liquid product distribution of converted $CO_2$ gas from oceanwater as a function of time using **a** Cu-catalyst-based and **b** Ag-catalyst-based vapor-fed $CO_2R$ cell.

magnetron sputtering, and the Ag catalyst was deposited on carbon-based GDL (Ion Power, Sigracet 29 BC) by drop casting Ag nanoparticles[43]. A traditional three-electrode configuration with the $CO_2R$ catalysts as the working electrode, Pt mesh as the counter electrode, and Ag/AgCl (1 M KCl) as the reference electrode in the anolyte reservoir, was used[44]. Bulk electrolysis was performed at an applied potential of −1.14 V vs. the reversible hydrogen electrode (RHE) for 2 h with an average $CO_2$ flow rate of 2.2 sccm from the BPMED system and the anolyte flow rate of 5 ml min$^{-1}$. The resulting current densities ranged from 53 to 77 mA cm$^{-2}$ (Supplementary Fig. 18). During the bulk electrolysis, 0.5 ml aliquots of electrolyte was taken every 10 min to monitor the liquid product distribution over time. As shown in Fig. 5a, using the Cu catalyst, 73% of the electrons were selective toward $CO_2R$ products, while ~20% went toward HER. The remaining 7% of electrons were lost due to the re-oxidation of liquid products to $CO_2$ in the anode chamber, or from being absorbed in the anion exchange membrane (AEM) and GDL[44]. The product distributions for both liquid and gas products are relatively stable during the 2 h bulk electrolysis with an average single pass conversion rate of ~6.7% for Cu. A range of reduction products were obtained using the Cu catalyst, and producing a single desired product either by improving the selectivity of the reaction, or by downstream product separation, is a key area of $CO_2R$, which is currently under intense research and development. Ag electrodes have been used for the selective reduction of $CO_2$ to CO at high FE in both aqueous and gas diffusion configurations[43,45–48]. As shown in Fig. 5b, at −0.6 V vs. RHE, the FE toward CO increased to >90% for a large portion of the bulk electrolysis, with HER as low as 5%, and an average single pass conversion rate of ~9.7%. The operating current density of the Ag-based vapor-fed $CO_2R$ cell ranged from 7.6 to 11.7 mA cm$^{-2}$ (Supplementary Fig. 19). The selectivity of CO dropped, while HER increased with time due to flooding of the GDL. In a separate experiment, we have tested a $CO_2$ feed directly from the outlet of the BPMED with $O_2$ present using a Ag-based $CO_2R$ cell and found that more than 80% of the electrons went toward ORR (Supplementary Fig. 20). As a result, $CO_2$ gas feeds free of $O_2$ impurities are important for high $CO_2R$ selectivity in lieu of the parasitic loses from ORR.

To operationally match the $CO_2$ flux in the BPMED and the vapor-fed $CO_2R$ cell, the ratio between the $CO_2$ reduction current density and the $CO_2$ capture current density was estimated to be 4, assuming and a 6-electron process with 100% $CO_2$ utilization and selectivity (Supplementary Note 6). For example, if the vapor-fed $CO_2$ cell operates at 300 mA cm$^{-2}$, the areal-matched BPMED should operate at ~75 mA cm$^{-2}$. Our cost analysis indicated that there is a trade-off between the cost of the electrochemical energy consumption and the membrane cost, and the BPMED is

cost-effective when operates at current density of >60 mA cm$^{-2}$ (Supplementary Note 5 and Supplementary Fig. 14). The successful coupling of the electrochemical $CO_2R$ cell and the $CO_2$ capture unit showed that $CO_2$ captured from oceanwater could be a potential carbon feedstock for renewable generation of fuels or chemicals.

**Discussion**

The energy penalties related to oceanwater pumping, gas stripping, and cooling for the implementation of large-scale ocean capture are calculated in Supplementary Note 7. In a land-based stand-alone system, where the oceanwater must be collected from open ocean, the energy for oceanwater intake, pre-treatment and pumping accounts for the majority of the energy penalty, and the total energy consumption excluding the BPMED system was calculated to be 4.6 kWh kg$^{-1}$ $CO_2$. By co-locating the ocean capture unit with a water desalination plant, the energy consumption excluding the BPMED can be reduced to 0.075 kW h kg$^{-1}$ $CO_2$. The electrochemical energy consumption for the BPMED operating at industrial scale current density was estimated to be 1.22 kWh kg$^{-1}$ $CO_2$ (Supplementary Note 7). To put these energy penalties into perspective, a DAC industrial plant requires a total energy ranging from 1.54 to 2.45 kWh kg$^{-1}$ of $CO_2$[8]. As a result, a co-locating ocean capture plant with a total estimated energy of 1.30 kWh kg$^{-1}$ $CO_2$ is energetically favorable over the DAC.

To evaluate the viability of the proof-of-concept system, a techno-economic analysis (TEA) based on the discounted cash flow method was carried out (Supplementary Note 8). In principle, process intensification such as co-location of the $CO_2$ capture plant with water desalination plant would lead to a major cost saving both in the capital and operational expenditures (CapEx and OpEx)[19]. Our TEA yielded levelized costs in the range from $0.5 to $0.54 kg$^{-1}$ $CO_2$ for a system that is co-located with a desalination plant and levelized costs between $1.87 and $2.05 kg$^{-1}$ $CO_2$ for a stand-alone system, which are close to previously reported values for similar systems[19]. While the current cost of ocean capture is significantly higher than $CO_2$ capture from point sources (between $0.06 and $0.08 kg$^{-1}$ $CO_2$)[49] and DAC (between $0.094 and $0.232 kg$^{-1}$ $CO_2$)[8], very little research and development has been devoted in this area and some cost estimates are still debatable for large-scale deployment. For example, the capital expenditure (CapEx) breakdown of the co-located system (Supplementary Fig. 22) showed that the membrane contactor constituted ~60% of the total equipment cost. Cost reduction on the current membrane contactor or potentially eliminating the membrane contactor unit by direct conversion of dissolved $CO_2$ in the oceanwater would significantly reduce the overall cost of the system. In addition, as described in Supplementary Note 10, while only <0.36 V is required to produce the

acidified stream and basified stream at the optimal pH, most reported BPMED system including this work needed more than 0.6 V to operate at even relatively low current densities. Improvement of water dissociation rates and lower the voltage requirements within the BPMED system would also improve the overall cost of the $CO_2$ capture from oceanwater.

In summary, we demonstrated a direct coupled, proof-of-concept electrochemical system that used a BPMED cell and a vapor-fed $CO_2R$ cell for electrochemical capture and conversion of $CO_2$ from oceanwater. Our BPMED replaced the commonly used water-splitting reaction with one electron, reversible redox reactions at the electrodes, and exhibited a record low electrochemical energy consumption of 155.4 kJ mol$^{-1}$ or 0.98 kWh kg$^{-1}$ of $CO_2$. The pH of the acidified stream from BPMED was optimized at 4.7 by controlling the operating current density and the input ocean-water flow rate to yield the lowest electrochemical energy consumption and the highest $CO_2$ output rate. The acidified stream was passed through three series connected membrane contactors, in which gaseous $CO_2$ was captured at an efficiency of 71%. Both experimental measurements and multi-physics modeling results showed that the voltage drop across the BPM during operation deviated significantly from the ideal voltage requirements, and the slow water dissociation kinetics at the BPM interface accounted for the majority of the cell voltage loss. The new BPMED cell design can be employed and scaled up both in a single stack configuration or a multi-stack configuration without introduction of any unintended chemical reactions. The captured $CO_2$ was then directly feed to a series of connected vapor-fed cells for electrochemical reduction of $CO_2$. The Cu-catalyst-based vapor-fed cell exhibited a total operating current density of 58 mA cm$^{-2}$ and an FE for $CO_2R$ to gas and liquid products of up to 73% at −1.14 V vs. RHE, while the Ag-catalyst-based vapor-fed cell showed an operating current density of 11.15 mA cm$^{-2}$ and an FE for $CO_2$ conversion to CO of up to 95% at −0.6 V vs. RHE. The proof-of-concept system provides a unique technological pathway for $CO_2$ capture and conversion from oceanwater with only electrochemical processes.

## Methods

**Chemicals**. All chemicals were used as received. Instant ocean® sea salt (Instant Ocean), potassium ferricyanide ($K_3[Fe(CN)_6]$, certified ACS crystalline, Fisher Chemical), potassium ferrocyanide trihydrate ($K_4[Fe(CN)_6]\cdot3H_2O$, 98.5–102.0%, crystals, AR® ACS, Macron Fine Chemicals™), sodium sulfate ($Na_2SO_4$, reagent plus®, ≥99.0%, Sigma-Aldrich), potassium bicarbonate ($KHCO_3$, BioUltra, ≥99.5%, Sigma-Aldrich), and potassium hydroxide (KOH, reagent grade, VWR).

**Preparation of electrodes**. The electrodes for the BPM electrodialysis unit were titanium (Ti) plates with a platinum (Pt) coating. Pt was deposited onto Ti plates (0.89 mm thick, annealed, 99.7% metal basis, Alfa Aesar) using AJA radio frequency (rf) magnetron sputtering from a Pt target (Kurt J. Lesker, 99.95%, 2-in. diameter). The argon (Ar) flow was kept at 20 sccm and the working pressure was held at 5 μbar. The rf power was 100 W and the deposition rate was ~0.667 Å s$^{-1}$. The deposition time was set to 25 min, and the thickness of the resulting Pt film was ~100 nm.

Electrochemical reduction of $O_2$ (ORR) was performed in a custom vapor-fed gas diffusion electrode (GDE) cell using a drop-casted Ag catalyst. The Ag-GDE was fabricated using previously reported procedures[43]. Briefly, a solution of commercially available Ag nanoparticles were suspended in methanol by sonication for 30 min. One hundred fifty microliters of this solution was drop-casted onto the GDL (Ion Power, Sigracet 29 BC) and allowed to dry at 200 °C in air for 1 h. After cooling to room temperature, the Ag-GDE was ready for use.

Electrochemical reduction of $CO_2$ ($CO_2R$) was performed in a custom vapor-fed GDE cell using a 200 nm layer of Cu or drop-casted Ag catalyst. The Cu catalyst was deposited on a gas diffusion layer (PTFE, Sterlitech) by rf magnetron sputtering from a Cu target (Kurt J. Lesker 99.95%, 2-in. diamater). The sputtering conditions for Cu were the same as those for Pt deposition. The deposition rate was ~0.556 Å s$^{-1}$ and the deposition time was 1 h, resulting a Cu film with an approximate thickness of 200 nm.

**Fabrication of the BPMED cell**. The electrodialysis unit was a home-built single stack cell that consisted of four compartments; an acidified compartment, a basified

compartment, a catholyte compartment and an anolyte compartment, enclosed with 1.5-cm-thick acrylic plates. The solution compartments were made by ethylene propylene diene monomer rubber (EPDM) rubber sheets. These compartments were filled with 1.5-mm-thick polyethylene mesh and were sealed against leaks using axial pressure. The catholyte and anolyte compartments were identical and had a spacing of 5 mm, while the acidified and basified compartments had a spacing of 2 mm. The acidified and the basified oceanwater compartments were separated by a Fumasep bipolar membrane (BPM, FuMa-Tech), and each electrolyte compartment was separated from the oceanwater compartment by a CEM (Nafion™ N324, Teflon™ Fabric Reinforced, Ion Power). The cathode, anode, CEMs, and BPM had the same active area of 64 cm$^2$.

**Electrodialysis measurement**. The input oceanwater solutions used in the experiments were synthetic oceanwater prepared by adding 35.95 g of Instant Ocean® sea salt per liter of deionized water. The oceanwater was loaded into the acidified compartment using a peristaltic pump (Simply Pumps™ Perimax PMP200) and the acidified stream was first collected in a reservoir container before being supplied to the basified compartment using another peristaltic pump (Simply Pumps™ Perimax PMP200). The flow rates of the acidified and basified streams were kept the same to ensure pressure balance in all oceanwater compartments of the BPMED cell. The output basified stream was disposed of to a waste collection bucket. The electrolyte was supplied to the electrode compartments using a peristaltic pump (Cole-Parmer Masterflex L/S) that was split with a T-junction to separate streams of catholyte and anolyte. The outputs of the electrode compartments were collected in the same reservoir to allow mixing and continuous recycling of the electrode solution. A constant electrical current was applied to the BPMED cell using a Keithley 2400 source meter (Tektronix®), and the voltage was recorded using I-V software (developed by Michael Kelzenberg, Caltech). The pH of the oceanwater was measured using a pH meter (Denver Instrument UB-10).

BPM voltage measurement was performed in a separate cell that consisted of the same number of compartments and configuration but with a membrane and electrode active area of 4 cm$^2$ and a spacing of 1 cm for each compartment. BPM voltage was determined by measuring the voltage difference between two Ag/AgCl reference electrodes (1 M KCl, CH instruments), each was placed in the acidified and basified compartment, while applying electrical current to the electrodes using a Keithley 2400 source meter (Tektronix®) in a four-wire sensing mode. The ohmic resistance in the solution was determined separately using electrochemical impedance spectroscopy.

**CO$_2$ capture measurement**. The dissolved $CO_2$ in the acidified stream of oceanwater was captured using a vacuum pump (Ulvac DTC-120E) through a series of membrane contactors (3 M™ Liqui-Cel™ MM-0.5×1), each with a maximum operating liquid flow rate of 30 ml min$^{-1}$. Prior to entering the acidified compartment, the fresh oceanwater was degassed using another vacuum pump (Edwards RV3) to remove atmospheric gasses dissolved in oceanwater. Between the membrane contactors and the vacuum pump, a home-built cold trap surrounded by dry ice was used to condense moisture from the output gas stream. The flow of the output gas was monitored using a mass flow meter customized from Alicat Scientific and was logged using Flow Vision SC™ software. The output gas was diluted with 88 sccm $N_2$ in a mixing chamber with a volume of 25 ml, and the composition of the gas mixture was characterized using a gas chromatograph (GC, Model 8610C) customized from SRI Instruments.

**Construction of the vapor-fed CO$_2R$ cell**. Two polyether ether ketone (PEEK) plates served as the cathode and anode compartment of the GDE cell. A Pt mesh and Cu-GDE were mechanically pressed against opposite sides of an AEM (Fumatech, FAA-3-50) to form the membrane electrode assembly between the two PEEK plates, which were screwed together. To contact the electrodes to the potentiostat two thin wire leads, one Pt and one Ti, were introduced to the backside of the Pt mesh and Cu-GDE, respectively, and connected to a potentiostat for electrochemical measurements (SP300, Bio-Logic). For the $CO_2R$ cell, the anode chamber contained 1.0 M KOH aqueous electrolyte for OER at a flow rate of 5 ml min$^{-1}$, while in the cathode chamber $CO_2$ gas was introduced at a flow rate of 1.5 sccm from the outlet of the ORR cell.

The ORR pre-electrolysis cell used to mitigate $O_2$ flow into the $CO_2R$ GDE cell used a Ag-GDE. This cell was assembled in the same manner as the $CO_2R$ cell. The ORR cell was used in a three-electrode configuration using Ag-GDE as the working electrode, Pt mesh as the counter electrode, and Ag/AgCl (1 M KCl) as the reference electrode (CH Instruments). The anode chamber contained a 1.0 M $KHCO_3$ aqueous electrolyte for OER at a flow rate of 5 ml min$^{-1}$. A potential of −0.6 V vs. RHE was applied to the Ag-GDE which was found to be the minimum potential to operate in a mass transport limited region for ORR as determined by linear sweep voltammetry (Supplementary Fig. 21).

**Product analysis**. Gas products were measured with online GC (Model 8610C) customized from SRI instruments. A thermal conductivity detector (TCD) was used to detect $H_2$, $O_2$, $N_2$, and CO, while a flame ionization detector (FID) was used to detect $CH_4$, and $C_2H_4$ products. A parallel column configuration using a

Molsieve 5 A column was employed for $H_2$, $O_2$, $N_2$, and CO separation, while a Haysep 5D column is used to separate $CH_4$, CO, $CO_2$, $C_2H_4$, and $C_2H_6$. An isothermal method was used with an oven temperature of 90 °C, TCD temperature 120 °C, FID temperature 390 °C, and injection valve temperature 60 °C. Ar carrier gas was set to 20 psi, $H_2$ methanizer gas set to 20 psi, and air pump set to 5 psi.

FEs were calculated using the equation:

$$FE = \frac{nF\chi F_m}{I},$$ (5)

where $n$ is the number of electrons required for a specific product (i.e. 2 for CO, 8 for $CH_4$ etc.), $F$ is Faraday's constant (96,485 C mol$^{-1}$ e), $\chi$ is the product fraction, $I$ is the current (A), and $F_m$ is the molar flow (mol s$^{-1}$) defined by:

$$F_m = \frac{PF_v}{RT},$$ (6)

where $P$ is pressure (atm), $F_v$ is the volumetric flow rate (L min$^{-1}$), $R$ is the gas constant (0.08205 L atm mol$^{-1}$ K$^{-1}$) and $T$ is temperature (K).

Liquid products were analyzed from the anode side of the cell with high-performance liquid chromatography (HPLC, Dionex UltiMate 3000) every 10 min. The eluent was 1 mM $H_2SO_4$ in water with a flow rate of 0.6 ml min$^{-1}$ and column pressure of 76 bar. The column was an Aminex HPX 87-H from Biorad, held at 60 °C with an internal heater. The detector was a UV detector set to 250 nm. Injection volume was 10 μL. FEs of liquid products were calculated using the equation:

$$FE = \frac{nFz}{Q},$$ (7)

where $z$ is the moles of product measured and $Q$ is the total charge passed (C) at the time of sampling.

## Data availability
The data that support the findings of this study are available from the corresponding author upon reasonable request.

## Code availability
All code used in simulations supporting this article is available from M.L.

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

## Acknowledgements

This material is based on work performed by the Joint Center for Artificial Photosynthesis, a DOE Energy Innovation Hub, supported through the Office of Science of the U.S. Department of Energy under Award Number DE-SC0004993. M.L. acknowledges support from the Swiss National Science Foundation through the Early Postdoc Mobility Fellowship, Grant P2ELP2_178290. The authors thank the support from Sempra Energy on the cost and energy analysis of $CO_2$ capture from oceanwater. The authors are grateful to Dr. Nathan F. Dalleska at Caltech for the ion chromatography characterization of the synthetic oceanwater.

## Author contributions

I.A.D., I.S. and C.X. developed the conceptual idea and designed the experiments. I.A.D. and I.S. executed the experiments. M.L. performed the multi-physics modeling and simulation. C.X. and H.A.A. advised and supervised the work. I.A.D., I.S., M.L., L.H., and C.X. interpreted the data and wrote the manuscript. W.-H.C. fabricated the Ag-GDE. All the authors contributed in the intellectual discussions and finalized the paper.

## Competing interests

The authors declare no competing interests.
