## [Peer Review File · Nature Communications]

Reviewers' Comments:

Reviewer #1:

Remarks to the Author:

The authors have addressed all my concerns and those of the other reviewers very thoroughly and to my satisfaction, and I recommend publication in Nature Communications. Before publication, I ask the authors to consider two additional comments:

1. It appears that you are flowing seawater through the BPMED system, but you do not mention scaling on the membranes due to divalent cations (Ca^{++} and Mg^{++}) in the seawater. I would be very surprised if you do not see scaling on the membranes. If you do not, it may only be because you have operated it for a short time. Please add a discussion of this to the manuscript. The way to avoid this is shown in the Eisaman et al. work published in IJGGC – you take a small (1% of the volume you actually decarbonize) volume of seawater, filter out the divalent cations, and run this pure brine (NaCl (aq)) through the BPMED unit to generate acid and base. You then use this acid and base to dose a much larger (100x the volume of the sample used to make the acid and base) of seawater and perform the desired pH shifts. This has the advantage of reducing your pumping energy requirements because only this small seawater volume needs to be flowed through the pressure drop of the BPMED unit. In fact, if the pH shifts were performed directly in the ocean (imagine the BPMED unit on a ship or offshore platform with an expandable net excluding marine life temporarily from the region where the pH shifts would occur), the only seawater you would need to pump at all would be that 1% used to make the acid and base. If indeed your experiments did flow seawater through the BPMED unit, I suggest you: (1) explain whether or not you observed scaling from precipitation of divalent cations; (2) discuss how in a deployed system, you would instead just process a small volume as discussed above; and (3) include in your energy and cost analysis, what the pumping energy and cost would be in in-situ ocean case where you only need to pump this smaller volume.

2. In the supplementary information when referring to Eqs. 58 and 59, there appears to be some mix up with the numbering.

SIGNED: Matthew Eisaman

Reviewer #2:

Remarks to the Author:

Reviewer Comments for Manuscript Number: NCOMMS-20-20054-T: "Electrochemical Extraction and Conversion of CO_2 from Seawater"

I appreciate the article revisions made by Digdaya et al. in their article combining CO_2 capture and conversion from seawater. They have extended the impact of the work from my perspective, and made the conclusions broader than a purely experimental demonstration from the initial submission. I feel that after further modifications of their new content into broader conclusions for the prospects of CO_2 capture and conversion from seawater, this work could be published in Nature Communications.

The primary aspect missing for me, especially given the additional TEA work, is a strong conclusion on whether or not the authors feel this line of study is an energy or cost viable alternative to DAC. The data which has been presented indicates to this reviewer that even in the best of scenarios, a BPMED seawater extraction will likely not be energetically or cost favourable to the DAC process. The phrasing of the article, however, would seem to give the opposite opinion. This is particularly true in the first paragraph of the intro. While I understand the need to motivate the reason for performing the work, I

feel the main conclusions in the TEA analysis are somewhat at odds with these initial motivations.

My additional comments for the manuscript based upon the revisions and comments by the other reviewers are below:

1. The authors address the energy penalties of the system in the new Supplementary Note 7. Here they find that the estimated energy cost of a co-located BPMED CO₂ capture system is 1.29 kWh kg⁻¹ CO₂, of which 1.22 kWh kg⁻¹ CO₂ comes from the BPMED itself assuming a current density of 200 mA/cm². This is then lower than the range of 1.53-2.45 kWh kg⁻¹ CO₂ for direct air capture. I have some questions and comments related to this:

a) Looking at the manuscript and SI, I'm confused about the author's assertion that the cell voltage in the case of 200 mA/cm² would be only 1.316 V, leading to an energy cost of 1.22 kWh kg⁻¹ CO₂ (SI page 22). In their experiments, for example, the cell voltage was shown to be 1.9 V at only 10 mA/cm². The authors note they use the calculated values in Figure S15, but this is only the BPM voltage, rather than the cell voltage. The latter of which is much higher due to necessary ohmic drops in the electrolyte.

Can the authors show examples in literature where the voltage is as low as described, or give rational why the assumed cell voltage can be so low? I would be curious how this would impact the TEA numbers as well.

b) From the authors' revision, it appears that without co-location of a CO₂ extraction process with a desalination plant, the BPMED extraction process is likely not viable from a cost or energy perspective vs. DAC. In this case it would then be nice at the end of the article to discuss some limitations of the approach as a CO₂ utilization technology. One option, for example, is to reference the maximum CO₂ reduction potential of a desalination co-located approach.

e.g. Given a current global desalination capacity of 120x10⁶ m³/day (43.8x10⁹ m³/year), and an assumed ten-fold increase in the coming years, then 0.0043x10⁹ tons CO₂/year (0.012% of annual global emissions) could be removed from indirect ocean capture co-located with desalination plants.

2. I feel the title could be less-generalized. I would rather have the specific technologies which were used be specified in the title. As phrased, the title seems at first more like a review, than a new research article.

3. I noticed that the authors gave a cost and energy comparison of all three technology options in their rebuttal to reviewer 1. Can this be added to the SI as well?

Reviewer #3:

Remarks to the Author:

I still do not believe that this article is of sufficient merit to be published in a high impact factor paper such as Nature Comms. The process described is from my perspective, still beyond the bounds of something that would realistically ever be deployed.

Based on my original comments, the authors have confirmed that the costs of this process in a stand alone arrangement are simply not viable. They now indicate that the process can work if it is co-located with a desalination plant. This allows them to ignore the SWIPP costs, substantially reducing the capex and opex. However, even then there are issues:

- a) A desalination plant run entirely on renewable energy needs to be found and be willing to provide the land footprint for this installation.
- b) This desalination plant needs to be willing to take the seawater downstream of the CO₂ extraction process, as the SWIPP costs need to be re-couped through the production of fresh water. However, this stream is now at pH of ~10.7. Can the materials of construction in a desalination plant withstand such a high pH, given the concurrent high salt concentration? I would be particularly concerned about the materials in the pump that increases the pressure to 50Bar.
- c) There could actually be some advantages in running the desalination RO process at such a high pH, as this will limit boron transfer to the fresh water stream. However, the fresh water permeate will now be at an even higher pH (hydroxide ions will cross the membranes in preference to any other anion). This means that the fresh water, for human consumption, needs to be pH adjusted. This eliminates the driving force for the overall process. That is, the water is not being returned to the ocean at a high pH to accept more CO₂. The pH adjustment could be made using CO₂, but this would require further capex and opex costs, either to generate a pure CO₂ stream or to process this water through a very large packed or bubble column to transfer this CO₂ from air (i.e. direct air capture).
- d) I do not think it will be feasible to return the remaining brine (retentate from the RO membranes) to the ocean with elevated pH, as the localised impact on biota will be too extreme. Desalination plants are already required to put in place extensive diffusion systems to reduce the impact of the elevated salt levels in the brine on ocean biota and I would not think that coupling this with high pH would ever be acceptable from an environmental perspective. It would again be necessary to add an additional step to reduce this pH, by contact with a CO₂ rich stream within a bubble or packed column. This step will again add additional capital cost as well as energy costs to move the water and the gas.
- e) The manuscript now states that 'An ion exchange unit may be added prior to the BPMED to remove or replace the Mg²⁺ and Ca²⁺ in the seawater.' It is simply not that easy to soften a stream containing as much NaCl as seawater. The addition of the ion exchange unit adds further costs and an additional disposal problem, as a concentrated brine or basic solution (more concentrated than seawater) would need to be used for ion exchange regeneration.
- f) Even before considering these additional costs, the article predicts a levelised cost of 0.5 to 0.54 \$ kg⁻¹ CO₂ which is significantly above that for direct air capture (between 0.1 and 0.23 \$ kg⁻¹ CO₂). The authors may argue that more R&D may bring these costs down, but this is also true for Direct Air capture. It will be very difficult to make up this much ground.

Reviewer #1 (Remarks to the Author):

The authors have addressed all my concerns and those of the other reviewers very thoroughly and to my satisfaction, and I recommend publication in Nature Communications. Before publication, I ask the authors to consider two additional comments: 1. It appears that you are flowing seawater through the BPMED system, but you do not mention scaling on the membranes due to divalent cations (Ca^{++} and Mg^{++}) in the seawater. I would be very surprised if you do not see scaling on the membranes. If you do not, it may only be because you have operated it for a short time. Please add a discussion of this to the manuscript. The way to avoid this is shown in the Eisaman et al. work published in IJGGC – you take a small (1% of the volume you actually decarbonize) volume of seawater, filter out the divalent cations, and run this pure brine (NaCl (aq)) through the BPMED unit to generate acid and base. You then use this acid and base to dose a much larger (100x the volume of the sample used to make the acid and base) of seawater and perform the desired pH shifts. This has the advantage of reducing your pumping energy requirements because only this small seawater volume needs to be flowed through the pressure drop of the BPMED unit. In fact, if the pH shifts were performed directly in the ocean (imagine the BPMED unit on a ship or offshore platform with an expandable net excluding marine life temporarily from the region where the pH shifts would occur), the only seawater you would need to pump at all would be that 1% used to make the acid and base. If indeed your experiments did flow seawater through the BPMED unit, I suggest you: (1) explain whether or not you observed scaling from precipitation of divalent cations; (2) discuss how in a deployed system, you would instead just process a small volume as discussed above; and (3) include in your energy and cost analysis, what the pumping energy and cost would be in in-situ ocean case where you only need to pump this smaller volume.

Response: The authors appreciate the reviewer's comments. Precipitation of divalent cations was observed in our experiments, and was described in the main text.

Page 19, line 10:

“However, experimental measurements using the synthetic seawater showed a pH of 8.5, close to the pre-acidified condition. This discrepancy was attributed to the presence of non-negligible amounts of Mg^{2+} and Ca^{2+} ions in the synthetic seawater (Supplementary Table 2), which preferentially reacted with OH^- and formed white precipitates of divalent hydroxides or carbonates, as observed during the experiments. In the absence of Mg^{2+} and Ca^{2+} , the basified stream would reach a pH of ~10.46 (Supplementary Note 5).”

and page 21, line 21:

“Softening the entire seawater volume using ion exchanger units would be cost prohibitive at the present scale. Alternatively, feeding the basified chamber with higher flow rate of seawater in relative to the acidified chamber and controlling the pH of the basified chamber so that a critical pH would never be reached in the basified chamber would minimize the divalent cation scaling in the system. Other membrane related anti-scaling materials and processes would also play an important role in the future deployment of this technology.”

Additional text has also been added to compare two systems as mentioned by the reviewer: one with direct seawater acidification, and the other with producing concentrate acid and base for seawater acidification.

Page 12, line 12:

“In this study, to remove CO₂ from seawater, the entire volume of the seawater needs to be pumped through the BPMED system, in which optimal pH was achieved by acidification. The entire electro dialysis process involved very mild pH swings. Alternatively, to reduce the SWIPP cost, a BPMED system that produces relatively concentrated acid (HCl) and base (NaOH) was proposed so that only a small fraction of the seawater needs to go through the BPMED system.^{1,2} However, the thermodynamic limit of the energy required for direct seawater acidification process was calculated to be much less than in the separate process where seawater is acidified and basified to HCl and NaOH, respectively (See Supplementary Note 10).”

2. In the supplementary information when referring to Eqs. 58 and 59, there appears to be some mix up with the numbering.

Response: The numbering in SI has been fixed.

SIGNED: Matthew Eisaman

Reviewer #2 (Remarks to the Author):

Reviewer Comments for Manuscript Number: NCOMMS-20-20054-T: “Electrochemical Extraction and Conversion of CO₂ from Seawater”

I appreciate the article revisions made by Digdaya et al. in their article combining CO₂ capture and conversion from seawater. They have extended the impact of the work from my perspective, and made the conclusions broader than a purely experimental demonstration from the initial submission. I feel that after further modifications of their new content into broader conclusions for the prospects of CO₂ capture and conversion from seawater, this work could be published in Nature Communications.

The primary aspect missing for me, especially given the additional TEA work, is a strong conclusion on whether or not the authors feel this line of study is an energy or cost viable alternative to DAC. The data which has been presented indicates to this reviewer that even in the best of scenarios, a BPMED seawater extraction will likely not be energetically or cost favourable to the DAC process. The phrasing of the article, however, would seem to give the opposite opinion. This is particularly true in the first paragraph of the intro. While I understand the need to motivate the reason for performing the work, I feel the main conclusions in the TEA analysis are somewhat at odds with these initial motivations.

Response: The introduction section has been revised to give a balanced view between DAC and seawater capture.

Page 2, line 20:

“Extraction of CO₂ from seawater also presents many challenges and limitations. For example, at the present scale the estimated cost of seawater intake, pre-treatment and pumping (SWIPP) in a land-based, stand-alone system is high, ~\$1.49 kg⁻¹ CO₂.²¹ The co-location with a desalination plant was proposed to reduce the SWIPP cost,²¹ however, the system scale for CO₂ removal would be limited to ~83 kt-CO₂ year⁻¹ based on the current largest desalination plant (production capacity of 1,039,000 m³ day⁻¹ desalinated water, assuming average recovery rate of 44%). Development of an off-shore, standalone system powered by renewables can alleviate competitive land use, allow unique access to off-shore CO₂ storage sites, and can provide a source of CO₂ for offshore enhanced oil recovery (EOR).”

My additional comments for the manuscript based upon the revisions and comments by the other reviewers are below:

1. The authors address the energy penalties of the system in the new Supplementary Note 7. Here they find that the estimated energy cost of a co-located BPMED CO₂ capture system is 1.29 kWh kg⁻¹ CO₂, of which 1.22 kWh kg⁻¹ CO₂ comes from the BPMED itself assuming a current density of 200 mA/cm². This is then lower than the range of 1.53-2.45 kWh kg⁻¹ CO₂ for direct air capture. I have some questions and comments related to this:

a) Looking at the manuscript and SI, I'm confused about the author's assertion that the cell voltage in the case of 200 mA/cm² would be only 1.316 V, leading to an energy cost of 1.22 kWh kg⁻¹ CO₂ (SI page 22). In their experiments, for example, the cell voltage was shown to be 1.9 V at only 10 mA/cm². The authors note they use the calculated values in Figure S15, but this is only the BPM voltage, rather than the cell voltage. The latter of which is much higher due to necessary ohmic drops in the electrolyte.

Can the authors show examples in literature where the voltage is as low as described, or give rational why the assumed cell voltage can be so low? I would be curious how this would impact the TEA numbers as well.

Response: The BPM voltage of 1.0 V at an operating current density of 200 mA cm⁻² is readily achieved.^{4,5} The ohmic loss does not contribute significantly to the overall voltage penalty, for instance, the ohmic voltage loss is ~50 mV at 200 mA cm⁻² assuming a 100 μm thick electrolyte of 0.5 NaCl. Hence, as concluded in the main text, the co-location ocean removal system is energetically favorable over DAC. While the detailed cost analysis is beyond the scope of this work, preliminary results indicates that at \$0.03/kWh, the operating voltage increase of an electro dialyzer, for example from 1.3 V to 2 V, resulted in a very small increase (4%) of the CO₂ extraction cost due to the cheap electricity in the OpEx.

b) From the authors' revision, it appears that without co-location of a CO₂ extraction process with a desalination plant, the BPMED extraction process is likely not viable from a cost or energy perspective vs. DAC. In this case it would then be nice at the end of the article to discuss some limitations of the approach as a CO₂ utilization technology. One option, for example, is to reference the maximum CO₂ reduction potential of a desalination co-located approach.

e.g. Given a current global desalination capacity of 120x10⁶ m³/day (43.8x10⁹ m³/year), and an assumed ten-fold increase in the coming years, then 0.0043x10⁹ tons CO₂/year (0.012% of annual global emissions) could be removed from indirect ocean capture co-located with desalination plants.

Response: The author appreciates the reviewer's comments, and completely agree with the reviewer on the scale of the co-location system. Additional text has been added in the introduction part to address that. While the current land-based stand-alone system does not offer cost or energy advantage over DAC, an off-shore, stand-alone system by minimizing the SWIPP cost can potentially compete with DAC with many unique advantages.

Page 2, line 20:

“Extraction of CO₂ from seawater also presents many challenges and limitations. For example, at the present scale the estimated cost of seawater intake, pre-treatment and pumping (SWIPP) in a land-based, stand-alone system is high, ~\$1.49 kg⁻¹ CO₂.²¹ The co-location with a desalination plant was proposed to reduce the SWIPP cost,²¹ however, the system scale for CO₂ removal would be limited to ~83 kt-CO₂ year⁻¹ based on the current largest desalination plant (production capacity

of 1,039,000 m³ day⁻¹ desalinated water, assuming average recovery rate of 44%). Development of an off-shore, standalone system powered by renewables can alleviate competitive land use, allow unique access to off-shore CO₂ storage sites, and can provide a source of CO₂ for offshore enhanced oil recovery (EOR).”

2. I feel the title could be less-generalized. I would rather have the specific technologies which were used be specified in the title. As phrased, the title seems at first more like a review, than a new research article.

Response: The authors appreciate the reviewer’s suggestions. However, we decided to keep the original title, as this work included several electrochemical techniques including BPMED and vapor-fed electrochemical cells, hence, electrochemical extraction and conversion CO₂ from seawater fits well with the scope of this work.

3. I noticed that the authors gave a cost and energy comparison of all three technology options in their rebuttal to reviewer 1. Can this be added to the SI as well?

Response: The cost discussion in response to Review#1 has been added in the main text, and the cost comparison has been add in Supplementary Note 8.

Reviewer #3 (Remarks to the Author):

I still do not believe that this article is of sufficient merit to be published in a high impact factor paper such as Nature Comms. The process described is from my perspective, still beyond the bounds of something that would realistically ever be deployed.

Based on my original comments, the authors have confirmed that the costs of this process in a stand alone arrangement are simply not viable. They now indicate that the process can work if it is co-located with a desalination plant. This allows them to ignore the SWIPP costs, substantially reducing the capex and opex. However, even then there are issues:

Response: From the current estimates, the land-based stand-alone system does not offer cost or energy advantage in relative to DAC, however, an off-shore, stand-alone system that eliminates the majority of the seawater intake cost can potentially compete with DAC with many unique aspects. An additional discussion paragraph has been added to address the reviewer's concern.

The authors also appreciate reviewer's detailed comments related to the co-location system and the technical concerns raised with coupling desalination plants. While we try to respond to these concerns as best as we can, the coupling between the demonstrated CO₂ removal system and a desalination plant is beyond the scope of this work. More importantly, the potential success of the oceanic CO₂ removal technology at large scale (1Gt/year) will be completely de-coupled from a desalination plant system.

Page 2, line 20:

“Extraction of CO₂ from seawater also presents many challenges and limitations. For example, at the present scale the estimated cost of seawater intake, pre-treatment and pumping (SWIPP) in a land-based, stand-alone system is high, ~\$1.49 kg⁻¹ CO₂.²¹ The co-location with a desalination plant was proposed to reduce the SWIPP cost,²¹ however, the system scale for CO₂ removal would be limited to ~83 kt-CO₂ year⁻¹ based on the current largest desalination plant (production capacity of 1,039,000 m³ day⁻¹ desalinated water, assuming average recovery rate of 44%). Development of an off-shore, standalone system powered by renewables can alleviate competitive land use, allow unique access to off-shore CO₂ storage sites, and can provide a source of CO₂ for offshore enhanced oil recovery (EOR).”

a) A desalination plant run entirely on renewable energy needs to be found and be willing to provide the land footprint for this installation.

Response: The future operation and deployment of desalination plant is out of the scope of this work. Future oceanic CO₂ removal system does not necessarily need to co-locate with a desalination plant.

b) This desalination plant needs to be willing to take the seawater downstream of the CO₂ extraction process, as the SWIPP costs need to be re-couped through the production of fresh water. However, this stream is now at pH of ~10.7. Can the materials of construction in a desalination

plant withstand such a high pH, given the concurrent high salt concentration? I would be particularly concerned about the materials in the pump that increases the pressure to 50Bar.

Response: The authors appreciate the reviewer's comment, however, the technical challenges related to the coupling between our demonstrated CO₂ removal system and a desalination plant is beyond the scope of this work. The authors have no expertise in the construction materials for desalination plants, and the implementation of the demonstrated CO₂ removal system may or may not involve a desalination plant.

c) There could actually be some advantages in running the desalination RO process at such a high pH, as this will limit boron transfer to the fresh water stream. However, the fresh water permeate will now be at an even higher pH (hydroxide ions will cross the membranes in preference to any other anion). This means that the fresh water, for human consumption, needs to be pH adjusted. This eliminates the driving force for the overall process. That is, the water is not being returned to the ocean at a high pH to accept more CO₂. The pH adjustment could be made using CO₂, but this would require further capex and opex costs, either to generate a pure CO₂ stream or to process this water through a very large packed or bubble column to transfer this CO₂ from air (i.e. direct air capture).

Response: The authors appreciate the reviewer's comment, however, the technical challenges related to the coupling between our demonstrated CO₂ removal system and a desalination plant is beyond the scope of this work. Furthermore, the implementation of the demonstrated CO₂ removal system may or may not involve a desalination plant.

d) I do not think it will be feasible to return the remaining brine (retentate from the RO membranes) to the ocean with elevated pH, as the localised impact on biota will be too extreme. Desalination plants are already required to put in place extensive diffusion systems to reduce the impact of the elevated salt levels in the brine on ocean biota and I would not think that coupling this with high pH would ever be acceptable from an environmental perspective. It would again be necessary to add an additional step to reduce this pH, by contact with a CO₂ rich stream within a bubble or packed column. This step will again add additional capital cost as well as energy costs to move the water and the gas.

Response: The author appreciates reviewer's comments on the environmental perspective. Additional text has been added to acknowledge the impact of returning decarbonized seawater.

Page 19, line 20:

“Note that the environmental impacts of returning decarbonized seawater at pH >10 with the same salt level is not well understood presently and subsequent processes will need to be developed and implemented to levitate any impact on oceanic life.”

e) The manuscript now states that ‘An ion exchange unit may be added prior to the BPMED to remove or replace the Mg²⁺ and Ca²⁺ in the seawater.’ It is simply not that easy to soften a stream containing as much NaCl as seawater. The addition of the ion exchange unit adds further costs and

an additional disposal problem, as a concentrated brine or basic solution (more concentrated than seawater) would need to be used for ion exchange regeneration.

Response: The reviewer is correct that ion exchange units add further costs. Additional text has been added in the manuscript to discuss this challenge:

Page 21, line 19:

“The cost analysis presented herein only captures the cost of electrodialysis and gas stripping and ignores the cost for seawater pre-treatment to prevent scaling. Softening the entire seawater volume using ion exchanger units would be cost prohibitive at the present scale. Alternatively, feeding the basified chamber with higher flow rate of seawater in relative to the acidified chamber and controlling the pH of the basified chamber so that a critical pH would never be reached would minimize the divalent cation scaling in the system. Other membrane related anti-scaling materials and processes would also play an important role in the future deployment of this technology.”

f) Even before considering these additional costs, the article predicts a levelised cost of 0.5 to 0.54 \$ kg⁻¹ CO₂ which is significantly above that for direct air capture (between 0.1 and 0.23 \$ kg⁻¹ CO₂). The authors may argue that more R&D may bring these costs down, but this is also true for Direct Air capture. It will be very difficult to make up this much ground.

Response: As stated in the manuscript, CO₂ from point sources (between \$0.06 and \$0.08 kg⁻¹ CO₂) and from DAC (between \$0.094 and \$0.232 kg⁻¹ CO₂) presents cost advantage in the present time in relative to oceanic CO₂ removal. However, oceanic CO₂ removal using off-shore, standalone system powered by renewables can potentially compete with DAC and can offer unique attributes relative to DAC, for example, the off-shore operation limits competitive land usage, allow unique access to off-shore CO₂ storage sites, and can provide a source of CO₂ for offshore enhanced oil recovery. A detailed TEA analysis of a future CO₂ removal system is beyond the scope of this work. The focus of this study is to demonstrate a BPMED system with a unique cell architecture and a proof-of-concept coupling between CO₂ extraction and CO₂ conversion devices.

Reviewers' Comments:

Reviewer #2:

Remarks to the Author:

The authors have sufficiently addressed my technical and novelty concerns for the article through their revisions, and I find this work to be publishable in Nature Communications in its current form.

The work covers a relatively large proof-of-concept merging of two technologies at a lower technology readiness level, as well as provides a well thought out technoeconomic analysis for a technical paper. The work provides a broad overview noting the opportunities and limitations of such a combined system that will provide a good foundation for future work aimed at determining the final industrial viability of such an approach compared to other technologies.

Reviewer #2 (Remarks to the Author):

The authors have sufficiently addressed my technical and novelty concerns for the article through their revisions, and I find this work to be publishable in Nature Communications in its current form.

The work covers a relatively large proof-of-concept merging of two technologies at a lower technology readiness level, as well as provides a well thought out technoeconomic analysis for a technical paper. The work provides a broad overview noting the opportunities and limitations of such a combined system that will provide a good foundation for future work aimed at determining the final industrial viability of such an approach compared to other technologies.

Response: The authors thank the reviewer for this comment.